# Comparison between Some Phenotypic and Genotypic Methods for Assessment of Antimicrobial Resistance Trend of Bovine Mastitis *Staphylococcus aureus* Isolates from Bulgaria

**DOI:** 10.3390/vetsci9080401

**Published:** 2022-07-31

**Authors:** Nikolina Rusenova, Nasko Vasilev, Anton Rusenov, Aneliya Milanova, Ivo Sirakov

**Affiliations:** 1Department of Veterinary Microbiology, Infectious and Parasitic Diseases, Faculty of Veterinary Medicine, Trakia University, 6000 Stara Zagora, Bulgaria; 2Department of Obstetrics, Reproduction and Reproductive Disorders, Faculty of Veterinary Medicine, Trakia University, 6000 Stara Zagora, Bulgaria; nasvas@abv.bg; 3Department of Internal Diseases, Faculty of Veterinary Medicine, Trakia University, 6000 Stara Zagora, Bulgaria; vetroussenov@abv.bg; 4Department of Pharmacology, Animal Physiology and Physiological Chemistry, Faculty of Veterinary Medicine, Trakia University, 6000 Stara Zagora, Bulgaria; akmilanova@gmail.com; 5Department of Medical Microbiology, Medical University of Sofia, Faculty of Medicine, 1431 Sofia, Bulgaria; insirakov@gmail.com

**Keywords:** *Staphylococcus aureus*, mastitis, cows, antimicrobial resistance, resistance genes

## Abstract

**Simple Summary:**

The aim of the study was to assess the resistance of bovine mastitis *Staphylococcus aureus* isolates to antimicrobials by comparison of laboratory methods and statistical analysis. For this, 546 mastitis milk samples at 14 farms from 9 districts in Bulgaria were examined. A total of 92 *Staphylococcus aureus* strains were isolated and identified. The results showed high resistance to sulfadimethoxine (87%), followed by resistance to penicillin (33.7%), erythromycin (13%), streptomycin (8.7%), tetracycline (6.5%) and gentamicin (1.1%). The comparison of the methods demonstrated more than 90% agreement for 9 tested antimicrobial drugs, hence proving reliability of results from monitoring of resistance. Considering the detected discrepancies for some of isolates, an integral evaluation through standard and modern molecular methods for *Staphylococcus aureus* is recommended. The results from this study add to the microbiology laboratory experience and strategies for mastitis prevention and control programmes.

**Abstract:**

The aim of this study was to assess the resistance of bovine mastitis *S. aureus* isolates from farms in Bulgaria to different classes of chemotherapeutic drugs by comparison of some phenotypic and genotypic methods by means of Cohen’s kappa statistics. The study comprised 546 milk samples from subclinical and clinical mastitis at 14 farms from 9 districts in the country. A total of 92 *Staphylococcus aureus* strains were isolated from tested samples and identified by *nuc* PCR. The results demonstrated high levels of resistance to sulfadimethoxine (87%), followed by resistance to penicillin (33.7%), erythromycin (13%), streptomycin (8.7%), tetracycline (6.5%) and gentamicin (1.1%). The comparison of both phenotypic tests with respect to 9 antimicrobials revealed strong agreement with kappa coefficient 0.836. An almost complete agreement was evidenced between phenotypic resistance to penicillin and *blaZ* gene presence, to methicillin with *mecA* gene, to tetracycline with *tet* genes, but the agreement between erythromycin resistance and *erm* genes presence was moderate. This study was the first to demonstrate discrepancy between the behaviour to cefoxitin in the disk diffusion test and oxacillin in the MIC test for an isolate shown to carry the *mecA* gene in the subsequent genetic analysis. Considering the detected discrepancies for some of isolates, an integral evaluation through phenotypic and molecular methods for monitoring of antimicrobial resistance of *Staphylococcus aureus* is recommended.

## 1. Introduction

*Staphylococcus aureus* (*S. aureus*) is among the most commonly isolated bacterial pathogens causing mild to severe and life-threatening infections in men [1,2]. *S. aureus* is described as agent of various infections in many domestic and wild animal species that may be transmitted to humans [3]. In cows, it is determined as causative agent of about one-third of udder inflammations–the main global health concern in dairy cattle farming [4]. Regardless of mastitis type (subclinical, clinical or chronic), farmers incur serious economic losses. The costs differ according to mastitis agent and lactation period [5]. For example, coagulase-negative staphylococci considered as minor pathogens cause no or low losses from milk yield compared to mastitis caused by *S. aureus* or *Escherichia coli* [5,6]. The progress in modern veterinary medicine and the new generation of diagnostic techniques assign great importance to intestinal microbiota and its protective role against development of *S. aureus* mastitis [7]. The milk microbiota, its interaction with microbial pathogens and possible effects on mastitis course are also investigated [8,9].

Apart the financial aspect of bovine udder inflammations, their public health impact is not less important. Non-prudent use of antibiotics for treatment and control of mastitis results in higher risk from selection of resistant *S. aureus*, including multiresistant ones, that may spread along the food chain [10,11,12]. Emergence of allergies and dysbacteriosis are other possible negative effects [13]. The detection of methicillin-resistant *S. aureus* (MRSA) isolates from cows with mastitis is especially alarming, as they often carry genetic determinants of resistance to antimicrobial drug classes other than beta-lactams [12,14,15,16,17]. Furthermore, the infections caused by these strains are refractory [18]. Several studies reported also the presence of methicillin-sensitive *S. aureus* (MSSA) resistant to erythromycin, tetracyclines, aminoglycosides, used in cattle farming for treatment of various infections, including intramammary ones [12,19,20]. The studies investigating the sensitivity of bovine mastitis *S. aureus* in Bulgaria at the phenotypic level are few, moreover data on the prevalence of genetic determinants of antibiotic resistance are not reported, which was the main incentive to undertake the present study.

*S. aureus* mechanisms of resistance to above mentioned antibiotics are well acknowledged. The *blaZ* gene, responsible for production of β-lactamase, is closely associated to emergence of resistance to penicillin [21]. Methicillin resistance is determined by the *mecA* gene, encoding a homologue of penicillin-binding protein PBP2 (PBP2a or PBP2’) whose affinity to β-lactams is low [22]. The resistance of erythromycin is due mainly to *erm* genes, and resistance to tetracyclines is encoded by *tet* genes [23,24].

The aim of this study was to assess the resistance of bovine mastitis *S. aureus* isolates from farms in Bulgaria to different classes of chemotherapeutic drugs by comparison of some phenotypic and genotypic methods by means of Cohen’s kappa statistics.

## 2. Materials and Methods

### 2.1. Sampling and Primary Identification of Staphylococcus *spp.* Isolates

A total of 546 milk samples from cows with subclinical and clinical mastitis were selected in 2019–2021 after screening for increased cell elements content with Mastitis test NK (Biovet, Ivanovice na Nané, Czech Republic) following manufacturer’s instructions. In addition to the increased somatic cell content in the tested quarters, inclusion criteria for sampling subclinical mastitis milk were both the absence of concurrent disease and dry period in the animal. Exclusion criterion for clinical mastitis sampling was the pre-treatment of the animal with either systemic or intramammary chemotherapeutic drug. From positive milk quarters, individual samples were collected for bacteriological examination in line with approved practices [25]. In the lab, 10 μL from each milk sample were routinely inoculated on tryptic soy agar (TSA, Sigma-Aldrich, Darmstadt, Germany) supplemented with 5% defibrinated sheep blood (TSBA), MacConkey agar (Himedia, Mumbai, India) for detection of Gram-negative bacterial microflora and mannitol salt phenol-red agar (Sigma-Aldrich) as a selective medium for presumptive identification of *S. aureus*. Plates were incubated aerobically for 24–48 h at 35 °C. Identification of *Staphylococcus* spp. was based on the absence of growth on McConkey agar, growth on blood agar, mannitol salt phenol-red agar and colonial characteristics of the isolates on the blood agar–size, morphology, presence of haemolysis. Fermentation of mannitol on the selective medium was also considered. After obtaining pure culture, Gram staining, catalase/oxidase tests and coagulase production in tube coagulase test with rabbit plasma (Hi Media, Mumbai, India) were carried out.

### 2.2. PCR Identification of S. aureus Isolates

DNA from presumptive *S. aureus* isolates was isolated with QIAmp DNA mini kit (Quiagen, Hilden, Germany) according to the protocol for Gram-positive bacteria. For cell wall lysis, the bacterial depot was treated with buffer containing 20 mg/mL lysozyme, 20 mM Tris HCl with pH 8.0, 2 mM EDTA and 1.2% Triton. DNA concentration was determined by UV-Vis spectrophotometry (Cary 60, Agilent Technologies, Santa Clara, CA, USA). Extracted DNA was stored at −20 °C until analysis. Isolates were identified through PCR amplification of species-specific sequences from the *nuc* gene of *S. aureus* with primers generating 359 bp amplicons [26]. The sequence of primers was as follows: au-F3 TCGCTTGCTATGATTGTGG, au-nucR GCCAATGTTCTACCATAGC. The reaction volume of 20 μL contained 0.25 μM of each primer, 1x PCR buffer, 1.5 mM MgCl_2_, 0.2 mM dNTPs, 1 U/rxn Taq polymerase (Canvax, Córdoba, Spain), 1 μL DNA. PCR temperature details are presented in Appendix A. PCR products were run in 2% agarose gel (Sigma-Aldrich, Burlington, MA, USA) stained with 10 ng/mL ethidium bromide (Sigma-Aldrich, Merck KGaA, Saint Louis, MO, USA).

### 2.3. Antimicrobial Susceptibility Testing

The sensitivity of *S. aureus* isolates to chemotherapeutic drugs at the phenotypic level was determined by the disk diffusion test as per CLSI [27] and by minimum inhibitory concentrations (MIC) using the serial dilutions micromethod with commercial plates (Trek Diagnostic Systems Ltd., East Grinstead, UK). The following antimicrobial drugs and concentrations were used in the former test: penicillin 10E, ampicillin 10 μg, amoxicillin/clavulanic acid 20/10, cefoxitin 30 μg, cephalothin 30 μg, ceftiofur 30 μg, streptomycin 10 μg, gentamicin 10 μg, tetracycline 30 μg, erythromycin 15 μg, enrofloxacin 5 μg, co-trimoxazole (sulfa/trimethoprim 23.75/1.25), penicillin/novobiocin 10/30 μg and pirlimycin 2 μg. The disks were provided by HiMedia (Mumbai, India), except for those containing penicillin, penicillin/novobiocin, ceftiofur and pirlimycin, purchased, respectively, from BB-NCIPD (Sofia, Bulgaria), BioLab (Budapest, Hungary) and Oxoid (Basingstoke, UK). Bacterial inocula from 18-20-h cultures of isolates on TSBA were prepared by the direct colony suspension method in physiological saline and brought to 0.5 McFarland standard (Densilameter II, Erba Lachema, Brno, CZ). Inoculated plates were incubated aerobically for 24 h at 35 °C. After measurement of growth inhibition zones, isolates were determined as sensitive, resistant and intermediate; the latter were considered as resistant from clinical point of view. A reference *S. aureus* ATCC 25923 strain was used as control.

Minimum inhibitory concentrations (MIC) were determined on Sensititre mastitis plate format CMV1AMAF with two-fold dilutions of selected antimicrobial drugs presented in Appendix A. Plates were inoculated with 50 μL standardised inoculum prepared according to the former method and diluted in cation adjusted Mueller-Hinton broth to achieve a final concentration of approximately 5 × 10^5^ cfu/mL. After sealing with adhesive foil provided by the manufacturer, plates were aerobically incubated at 35°C for 24 h. After the incubation, results were read manually, considering first positive growth controls (3 wells) for each of isolates. MICs were interpreted as the lowest antimicrobial drug concentration inhibiting visible growth. The reference strains *S. aureus* ATCC 29213 and *Escherichia coli* ATCC 25922 were used as controls for each batch of plates. The interpretation of results from both phenotypic methods was carried out according to CLSI criteria [27]

### 2.4. Detection of Genes Encoding Resistance to Antibiotics

*S. aureus* isolates resistant and sensitive to beta-lactams, tetracyclines and erythromycin were analysed for presence of *mecA*, *blaZ*, *tetM*, *tetK*, *ermB* and *ermC* genes by means of conventional PCR. The sequences of primers expected PCR product size and their origin are listed in [19,28] Both singleplex and multiplex PCR reactions were performed. In the first variant, the PCR protocol described for *nuc* gene detection was followed. A multiresistant methicillin-resistant *S. aureus* DSM 29134 provided by the German Collection of Microorganisms and Cell Cultures GmbH served as positive control.

The multiplex reaction was performed with two primer pairs–*blaZ/ermC* and *blaZ/tetK.* The optimum hybridisation temperature of primers was selected through a gradient PCR within the range 49.9–56.1 °C. The reaction included 35 cycles with the following concentrations of reagents for the first primer pair: 0.5 μM of each primer, 1x PCR σyϕep, 2 mM MgCl_2_, 0.3 mM dNTPs, 1.5 U/rxn Taq polymerase (Canvax, Spain), 4 μL DNA and molecular biology grade water (Sigma-Aldrich, USA) up to 20 μL. For the second primer pair, reagent concentrations were: 0.5 μM of each primer, 1x PCR σyϕep, 1.5 mM MgCl_2_, 0.2 mM dNTPs, 1 U/rxn Taq polymerase, 4 μL DNA and water up to 20 μL Temperature conditions and exposures comprised single denaturation at 95 °C for 5 min, denaturation at 95 °C for 30 cek, annealing at 55 °C for the first reaction mix and 52 °C for the second reaction mix for 35 s, elongation at 72 °C for 60 s and final amplification at 72 °C for 7 min.

Gel electrophoresis was performed with 2% agarose (Sigma-Aldrich, Burlington, MA, USA) stained with 10 ng/mL ethidium bromide (Sigma-Aldrich, USA), 1 × TBE buffer, DNA ladder (100 bp, BrightMax, Córdoba, Spain) at 120 V, 400 mA for 30 min. The gel was visualised using UV transilluminator (ImageQuant 150, GE Healthcare, Taipei, Taiwan) at 240/260 nm.

### 2.5. Statistical Analysis

The agreement of results from the behaviour of isolates against chemotherapeutic drugs by the disk diffusion method and MIC method were compared by Cohen’s kappa statistic and interpreted according to McHugh [29]. The Mueller-Hinton broth dilution micromethod was used as gold standard for comparisons [30]. The sensitivity and specificity of PCR tests for detection of genes encoding resistance to antibiotics and the corresponding behaviour at the phenotypic level were determined by formulas described by Martin [31]. Cohen’s kappa statistic was used to determine the agreement between phenotypic resistance and presence of genes of resistance to beta-lactams, erythromycin and tetracyclines. Statistically significant differences in the resistance against the tested antibacterial drugs in the investigated farms were assessed by ANOVA test, followed by Bonferoni post-hoc test. Statistycal evaluation of the results was performed with a specialized software (STATISTICA for Windows 10.0, StatSoft, Inc., Tulsa, Oklahoma, USA).

## 3. Results

### 3.1. Primary Identification of S. aureus Isolates

A total of 92 presumptive *S. aureus* strains were isolated from tested 546 milk samples from cows with subclinical and clinical mastitis. The results from primary identification tests of the staphylococcal isolates are presented in Appendix A. With the exception of 3 isolates, all demonstrated haemolytic activity on blood agar with sheep red blood cells: 63/68.5%-double haemolysis and 26/28.3%-beta haemolysis. Gram-stained microscopic preparations demonstrated Gram-positive cocci arranged in grape-like clusters. Tests for production of catalase and oxidase were positive and negative, respectively. The pigmentation of colonies, mannitol fermentation on mannitol-salt agar and coagulase production of isolates were variable. A major part of isolates produced colonies with yellowish tint –69/75%, the rest were not pigmented. Seven isolates gave a mild positive mannitol fermentation reaction (7.6%), and another 8 were negative in the free coagulase tube test (8.7%). The strains originated from 14 farms located in 9 administrative districts and 4 regions in Bulgaria (Appendix A).

### 3.2. PCR Identification of S. aureus Isolates

Presumptive *S. aureus* isolates were precisely identified at the species level by conventional PCR. The results demonstrated 359 bp amplification products, specific for the *nuc* gene of *S. aureus* in all tested isolates (Figure 1).

### 3.3. Antimicrobial Susceptibility Testing

In this case, 32 (34.8%) of the 92 *S. aureus* isolates tested by the disk diffusion method demonstrated resistance against selected antimicrobial drugs. Resistant isolates originated from 9 farms in 4 regions of the country. At the farm level, the resistance of *S. aureus* to tested antimicrobials varied from 0% to 75%. In 5/14 (35.7 %) of farms, resistance to none of used chemotherapeutics was detected. These farms were located in Pazardzhik. Stara Zagora, Sliven, Burgas and Kardzhali districts, situated in the South central and Southeastern regions of Bulgaria (Appendix A).

The highest, statistically significant, resistance rate was found out against penicillins-29/92 (31.5% of all isolates, *p* < 0.0001). There were not statistically significant differences in the resistance against the other tested antimicrobials between the farms. The resistance rate to erythromycin was 9/92 (9.8%, *p* > 0.05), to streptomycin–8/92 (8.7%, *p* > 0.05), to tetracyclines–6/92 (6.5%, *p* > 0.05), to gentamicin and ceftiofur–1/92 (1.1%, *p* > 0.05), respectively. *S. aureus* isolates resistant to the other tested antimicrobials drugs were not found out (Figure 2).

The resistance patterns of *S. aureus* isolates are presented in Appendix A and depicted in Appendix A. Most of isolates were resistant to a single chemotherapeutic drug class-3 isolates to streptomycin, 18 to penicillin (ampicillin) or to two antibiotic classes-3 isolates resistant to penicillin (ampicillin) and erythromycin and one: to penicillin (ampicillin) and tetracycline. Multiresistance to three and more groups of antimicrobial drugs was exhibited by 7 isolates (7.6%) and the following patterns were observed: penicillin (ampicillin), tetracycline and erythromycin (*n* = 2), (P (A)-T-E); penicillin (ampicillin), streptomycin and erythromycin (*n* = 2), (P (A)-S-E); penicillin (ampicillin), streptomycin, tetracycline and erythromycin (*n* = 2), (P (A)-S-T-E); 1 isolate was resistant to penicillin (ampicillin), ceftiofur, streptomycin, gentamicin and tetracycline (P (A)-EFT-S-G-T). Multiresistant isolates were isolated only from cows with clinical mastitis.

Data about minimum inhibitory concentrations of the 10 chemotherapeutic drugs in the mastitis pathogens plate for *S. aureus* are presented in Table 1.

In most isolates determined as sensitive in the disk diffusion test, MIC values of chemotherapeutics in the plate were within the reference ranges. There were isolates, whose growth was not inhibited by the highest concentration of an antimicrobial in the plate. For example, no inhibition was detected from penicillin, ampicillin, erythromycin, tetracycline and sulfadimethoxine in 6.5%, 5.4%, 8.7%, 6.5% and 87% of tested isolates. For them, respective MIC_90_ values were 2 µg/mL, 4 µg/mL, 1 µg/mL, 1 µg/mL and >256 µg/mL.

Table 2 presents the data from the general comparison between the results from the disk diffusion and MIC tests for 9 chemotherapeutic drugs against bovine mastitis *S. aureus* isolates. The agreement between both tests was 92.39% with kappa coefficient 0.836 [95% confidence interval (CI) 0.719–0.952; standard error (SE) of kappa–0.059].

With respect to penicillin (Appendix A) and ampicillin (Appendix A), data were similar: agreement 98.91%, kappa coefficient 0.975, 95% CI 0.925(6)–1.000, SE 0.025 (0.000).

The results for cefoxitin/oxacillin were the same as for ampicillin, and those for ceftiofur–the same as for penicillin. The comparison of tests with respect to erythromycin is shown in Appendix A. The results demonstrated 96.74% agreement with kappa coefficient 0.83(95% CI 0.662–1.000; SE of kappa 0.090).

For tetracycline (Appendix A), the tests agreed completely: 100%; kappa coefficient 1.000 (95% CI 1.000–1.000; SE 0.000). The results of cephalotin, penicillin/novobiocin and pirlimycin corresponded to those for tetracycline.

### 3.4. Detection of Resistance Genes

The singleplex PCR assays demonstrated amplification products with size 310 bp, 517 bp (Figure 3), 190 bp and 169 bp (Figure 4), corresponding to, *mecA*, *blaZ* genes and *ermC* and *tetK* genes, respectively, in 2% agarose gel ((Sigma-Aldrich, USA) stained with 10 ng/mL ethidium bromide (Sigma-Aldrich, USA).

Out of the 29 isolates resistant to penicillin, all possessed the *blaZ* gene. *BlaZ* gene was not detected in 61 out of all 63 penicillin-sensitive isolates. The sensitivity and specificity of PCR assay for detection of resistance to penicillin were 100% and 96.8%, respectively. Among the 12 isolates resistant to erythromycin, the *erm* gene was found out in 7, furthermore, it was detected in 2 of susceptible isolates; therefore, the sensitivity of PCR for detected of resistance to erythromycin was 58.3%, and specificity–97.5%. With regard to resistance to methicillin and tetracycline, both the sensitivity and specificity of the reaction were 100%.

The results from the subsequent multiplex PCR analysis corresponded entirely to those of reactions with single primer pairs (Figure 5).

The agreement between phenotypic resistance to penicillin and the presence of the *blaZ* gene was 97.83% with kappa coefficient 0.951 (95% CI 0.883–1.000; SE of kappa 0.035). The agreement between phenotypic and genotypic tests for erythromycin was 92.39% with kappa 0.625 (95% CI 0.371–0.878; SE of kappa 0.129); for methicillin and tetracycline–kappa 1.000; 100% agreement.

## 4. Discussion

The first step for characterisation of bacterial strains is isolation of pure culture and precise identification by means of contemporary methods with high sensitivity and specificity. The performed primary identification of bovine mastitis *S. aureus* isolates in this study showed variable results from key tests, e.g., presence of haemolysins, pigment production, mannitol fermentation, rabbit plasma coagulation, which was in line with previous observations of ours [32,33]. In a number of small-scale microbiological labs, the identification of *S. aureus* at the species level is based on the free coagulase tube test. It is not considered a definitive test, as coagulase-negative *S. aureus* strains are also isolated, probably due to mutations in the *coa* gene–an important epidemiological marker for detection of different *S. aureus* variants [34]. Thus, such *S. aureus* isolates may be erroneously classified as coagulase-negative staphylococci (CNS), which usually cause self-limiting udder infections [35]. The wrong diagnosis can influence the sufficiency of measures for mastitis control. On the other hand, coagulase-positive staphylococci other than *S. aureus* (*S. intermedius*, *S. pseudintermedius*, *S. shleiferi* subsp. *coagulans*) may be also involved in the etiology of mastitis [32,36,37]. In this study, all presumptive *S. aureus* isolates after basic biochemical tests were identified with conventional PCR based on the *nuc* gene, encoding thermonuclease, including isolates with atypical behaviour. The selected gene is strongly conservative and may be successfully used for precise identification of *S. aureus*, as confirmed by other research reports [19,26,38,39].

The testing of antimicrobial susceptibility of *S. aureus* isolates as well as of other causative agents is essential element of mastitis control strategies [40]. Selected antimicrobials comprised drugs used for treatment of mastitis in our country as well as preparations non-registered for use: pirlimycin or ceftiofur, erythromycin under the form of intramammary infusion, as well as penicillin/novobiocin to investigate the trends in antimicrobial resistance of this bacterial species, isolated from mastitic milk in Bulgaria. In this case, 14 substances were tested in the first method, MICs were determined for 10 antimicrobials, and comparisons of *S. aureus* behaviour in both phenotypic tests were made for 9 antimicrobial drugs included in the Sensititre mastitis plate format. The results determined 32 isolates (34.8%) as resistant to at least one of antimicrobial drugs by the disk diffusion test, whereas the MIC test identified 35 isolates (38%); therefore, a strong agreement between the two tests was found out with kappa value of 0.836. The resistance rates of isolates varied among the farms, districts and regions from 0% to 75%, which required a differentiated approach in programmes for mastitis prevention and control. For penicillin and ampicillin, both tests showed an almost perfect agreement with kappa coefficient 0.975. Disagreements referred to two different isolates. One isolate was interpreted as resistant by the disk diffusion test: 20 mm inhibition zone (IZ), but sensitive in the MIC test (0.12 µg/mL), and another isolate–sensitive to ampicillin by the disk diffusion test (39 mm IZ), but resistant by MIC (8 µg/mL). It should be noted that tests were performed under strictly controlled conditions to eliminate factors that would lead to imprecision in detection of results [27]; also, tests were repeated in independent experiments when inconsistent results for respective isolates were obtained. The paradoxical findings can be most probably due to the Eagle phenomenon–the ability of microorganisms to survive and replicate at concentrations higher than their optimum minimum bactericidal concentration–effect described for *S. aureus* against penicillin and other beta-lactams [41]. Discrepancies between the disk diffusion test and MIC determined by the E-test were reported by another study [42] having tested the sensitivity of *S. aureus* isolates from mammary glands of cows and sheep, the authors detected error rates of 3.3% for penicillin and 3.8% for ampicillin. Disagreement was demonstrated by Khalili et al. [43] in testing *S. aureus* and *Enterococcus* spp. susceptibility to beta-lactams, glycopeptide antibiotics and clindamycin. Before being reported as sensitive to penicillin, *S. aureus* strains should be tested for production of beta-lactamase [27]. That is why this study tested isolates with phenotypic resistance and those sensitive to penicillin for the presence of the *blaZ* gene. The results showed a high sensitivity and specificity of the PCR method for *blaZ* gene detection compared to phenotypic tests with almost perfect agreement (kappa coefficient 0.951), in line with data from other studies [19,44]. The established 33.7% resistance to penicillin after the genetic analysis (*n* = 31/92 *blaZ* positive isolates) was relatively low compared to reported rates from Estonia (61.4%), England (36 to 46%), Korea (38.6 to 78.8%), some farms in China (>60%); northern Ethiopia (>90%), southeastern Brazil (60.7%), Comarca Lagunera region of Mexico (97%) etc. [45,46,47,48,49,50,51]. Lower rates of resistance to penicillin were reported from Chinese dairy farms in Shandong, Jiangsu and Guangdong provinces–24.2% [12], farms in East Poland–23.6% [52], Switzerland–14% [53] and others. In our country, Nikolova et al. [54] have tested the susceptibility of *Staphylococcus* spp. isolates from two dairy cattle farms in northeastern Bulgaria against different antimicrobials. The authors identified 8 coagulase-positive staphylococci, which were 100% sensitive to tested antimicrobial drugs at farm 1 whereas at farm 2–18.2% were resistant to amoxicillin: a rate comparable to our results.

A surprising finding of this study was the detection of a methicillin-resistant *S. aureus* isolate from clinical bovine mastitis, which was initially clearly defined as sensitive to cefoxitin by the disk diffusion test on the basis of the 22 mm IZ. A zone of the same size was observed also with the MRSA strain DSM 29134, used as reference strain in the PCR assay, which was also atypical. Subsequent analyses demonstrated resistance to oxacillin with MIC > 4 µg/mL and presence of the *mecA* gene in both strains. To our best knowledge, this is the first report for isolation of MRSA from a cow with mastitis in Bulgaria, confirmed to carry the *mecA* gene. Additionally, this clinical isolate showed an unique resistance profile: to penicillin (ampicillin), ceftiofur, streptomycin, gentamicin and tetracycline by the disk diffusion test and susceptibility to amoxillin/clavulanic acid. It was determined as sensitive to erythromycin by both phenotypic tests, but according to the PCR assay, carried the *ermC* gene. Further detailed studies of this isolate at the molecular level are necessary to establish its SCCmec type as well as full genome sequencing for better understanding of antimicrobial resistance mechanisms. Oxacillin-sensitive *mecA* positive *S. aureus* are reported in the literature [55,56], yet the present study is the first to report disagreement between the behaviour to cefoxitin in the disk diffusion test (used as surrogate in oxacillin resistance testing) and that to oxacillin in the MIC test.

With respect to erythromycin, the agreement of results of both phenotypic tests was strong (kappa coefficient 0.839), yet the agreement of phenotypic test and the presence of *erm* genes was only moderate (kappa coefficient 0.625). From all 12 detected erythromycin-resistant isolates (9 in the disk diffusion and MIC tests; another 3 only in the MIC test), seven strains carried the *ermC* gene whereas the *ermB* gene was absent in either resistant or sensitive isolates. The observed sensitivity of the PCR method for detection of genes encoding resistance to erythromycin–58.3% was much lower than the 100% sensitivity reported by Gao et al. [19] for detection of *ermC/ermB* genes. This was probably due to the presence of *ermA* gene that was not tested in our isolates and is a limitation of this study. Nevertheless, the prevalence of *erm* genes in different research reports is highly variable. For example, Ghanbari et al. [23] observed the highest prevalence of *ermC* followed by *ermB* and *ermA* genes among tested human *S. aureus* isolates; in addition, they did not detect any *erm* gene in two strains with inducible MLS_B_ phenotype. Another study [57] reported 60% prevalence of the *ermA* gene among erythromycin-resistant *S. aureus* isolates. The interpretation of resistance to macrolides should be careful and integral as the different resistance mechanisms produce a different phenotypic profile [58]. At the farm level, the resistance to erythromycin varied from 0 to 46.2%, taking into account that in Bulgaria, this antibiotic is not applied as intramammary infusion. The common use of tylosin in veterinary medical practice may have resulted in horizontal transfer of genetic elements conferring cross resistance in *S. aureus* to this antimicrobial drug class [59]. In addition, resistance was confirmed in both subclinical and clinical mastitis isolates.

The results for the behaviour of isolates to tetracycline agreed almost perfectly between the two phenotypic tests, as well as with the presence of *tet* genes, among which only the *tetK* gene was found out. The resistance rate to tetracycline in all tested isolates was low (6.5%) compared to other reports [49,50,60]. In the different farms however, resistance rates varied from 0 to 50% which was not surprising at the background of the broad use of this antibiotic in cattle farming, including for treatment of mastitis. Rates from 0 to 30.8% in farms were obtained also with respect to streptomycin (in the disk diffusion test), used in combination with penicillin as intramammary infusion–an inevitable effect of the selective pressure of antimicrobial drugs on bacteria [18].

A perfect agreement on susceptibility of *S. aureus* with kappa value 1.000 was obtained from the two phenotypic tests for cephalothin, penicillin/novobiocin and pirlimycin. With regard to ceftiofur, included in Sensititre mastitis plate format, disagreement referred only to one isolate exhibiting intermediate sensitivity in the disk diffusion test and therefore classified as resistant, but then showing susceptibility with MIC = 2 µg/mL. Against sulfadimethoxine, the isolates from the present study showed high resistance rates: 87% were not inhibited at concentrations of 256 µg/mL, conversely to the behaviour of *S. aureus* strains in the study of Freu et al. [50] reporting 100% susceptibility. In this study, the sensitivity to amoxicillin/clavulanic acid, fluoroquinolones (enrofloxacin) and co-trimoxazole, tested by the disk diffusion method was 100%, contrary to other research data [39,49,61].

## 5. Conclusions

Bovine mastitis *S. aureus* isolates in the present study demonstrated preserved sensitivity to most of tested antimicrobial drug classes, high resistance to sulfadimethoxine and moderately high resistance to penicillinase-sensitive penicillin (33.7%). The comparative analysis between the disk diffusion test used in smaller veterinary diagnostic laboratories and MIC determination by the serial dilution micromethod applied in labs handling a greater number of samples showed more than 90% agreement for 9 tested antimicrobial drugs, hence proving reliability of results from phenotypic monitoring of resistance. By reason of observed discrepancies between the behaviour of some isolates in the phenotypic tests, e.g., against penicillin, ampicillin, cefoxitin/oxacillin and erythromycin, integral evaluation of resistance patterns with inclusion of molecular biological methods is advised. This approach resulted in detection of a cefoxitin-sensitive but oxacillin-resistant *mecA-*positive *S. aureus* isolate, as well as of few isolates sensitive in the disk diffusion test but resistant in the MIC test and vice versa. We believe that the results from this study will contribute to the experience of microbiological diagnostic labs and strategies implemented in mastitis prevention and control programmes.

## Figures and Tables

**Figure 1 vetsci-09-00401-f001:**
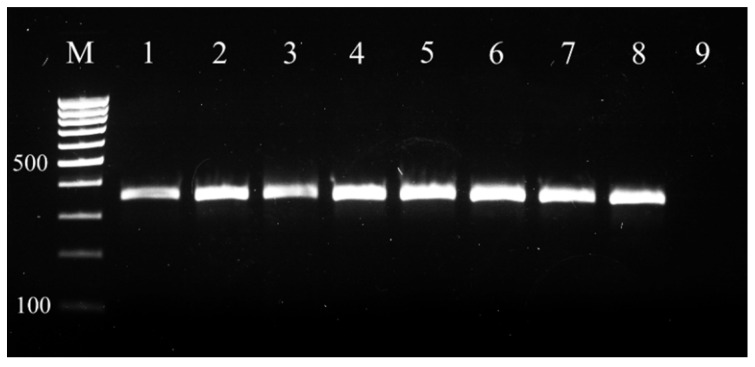
PCR identification of *S. aureus* isolates on the basis of *nuc* gene. M–100 bp molecular marker (BrightMax 100–1000 bp, Canvax, Spain); 1–positive control *S. aureus* ATCC 25923; 2-8–tested isolates; 9-negative control (water).

**Figure 2 vetsci-09-00401-f002:**
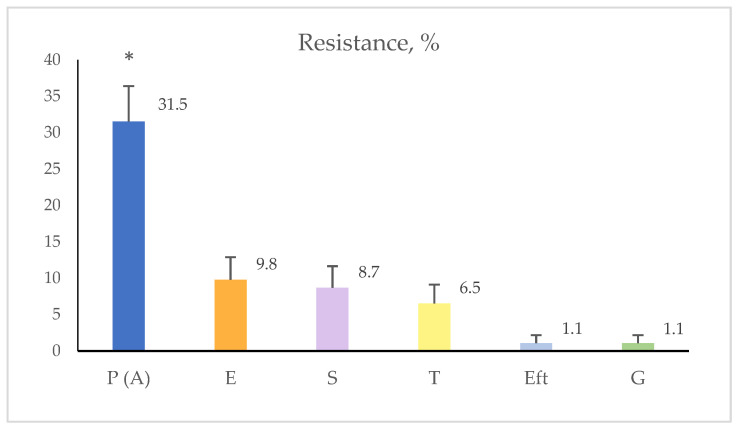
Percentage distribution of *S. aureus* isolates from bovine mastitis, resistant to tested chemotherapeutic drugs in the Bauer-Kirby disk diffusion test. **Legend:** P-penicillin, A-ampicillin, E-erythromycin, S-streptomycin, T-tetracycline, Eft-ceftiofur, G–gentamicin. * Statistically significantly higher resistance against penicillin in comparison to the other antibiotics in the investigated farms.

**Figure 3 vetsci-09-00401-f003:**
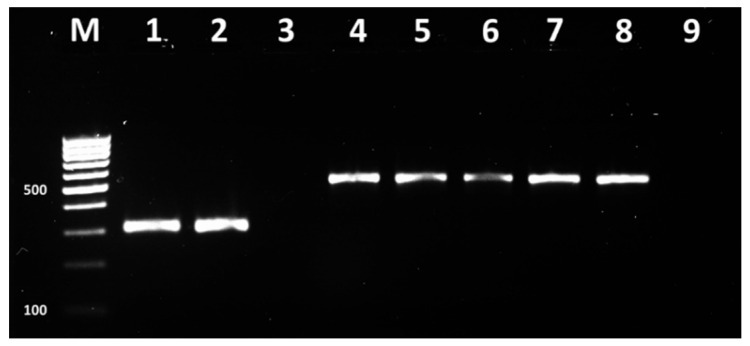
PCR demonstrating *mecA* and *blaZ* genes in bovine mastitis *S. aureus* isolates from Bulgaria. M–DNA ladder 100 bp (BrightMax 100–1000 bp, Canvax, Spain), 1–MRSA DSM 29134 *mecA* positive control, 2–*mecA* positive isolate, 3–negative control of first reaction mix; 4–MRSA DSM 29134 *blaZ* positive control, 5-8–*blaZ* positive isolates, 9–negative control of second reaction mix.

**Figure 4 vetsci-09-00401-f004:**
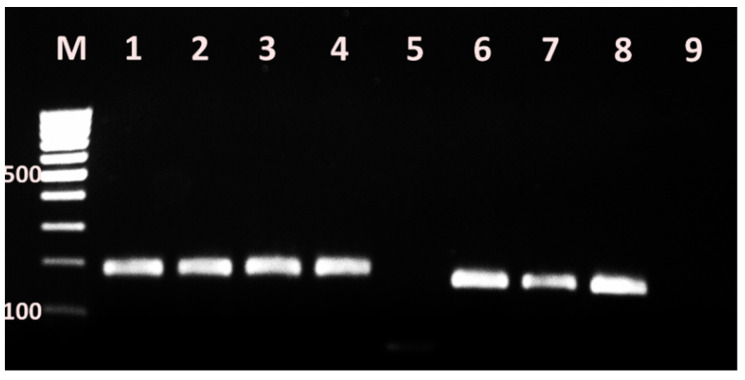
PCR demonstrating *ermC* and *tetK* genes in bovine mastitis *S. aureus* isolates from Bulgaria. M–DNA ladder 100 bp, 1–4–*ermC* positive isolates, 5-negative control of first reaction mix; 6–8–*tetK* positive isolates, 9-negative control of second reaction mix.

**Figure 5 vetsci-09-00401-f005:**
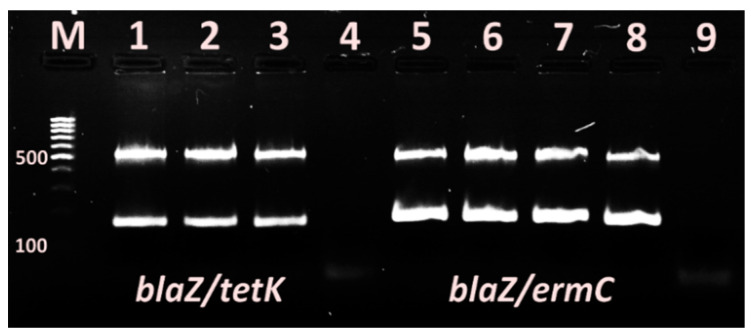
Multiplex PCR demonstrating the presence of *blaZ/tetK* and *blaZ/ermC* genes in some *S. aureus* isolates. 4, 9–negative controls of first and second reaction mix.

**Table 1 vetsci-09-00401-t001:** Distribution of minimum inhibitory concentrations of 10 antimicrobials included in Sensititre Mastitis plate format to *S. aureus* isolates from bovine mastitis in Bulgaria.

Antimicrobials	Distribution (%) of MIC (µg/mL)												NI	MIC_50_	MIC_90_
	0.12	0.25	0.5	1	2	4	8	16	32	64	128	256			
**PEN**	68.5	4.3	8.7	5.4	4.3	0.0	2.2	-	-	-	-	-	6.5	0.12	2
**AMP**	64.1	3.3	12.0	7.6	2.2	3.3	2.2	-	-	-	-	-	5.4	0.12	4
**ERY**	-	18.5	68.5	3.3	1.1	0.0	-	-	-	-	-	-	8.7	0.5	1
**OXA+**	-	-	-	-	98.9	1.1	-	-	-	-	-	-	-	2	2
**PIRL**	-	-	54.3	45.7	0.0	0.0	-	-	-	-	-	-	-	0.5	1
**TET**	-	-	-	90.2	3.3	0.0	0.0	-	-	-	-	-	6.5	1	1
**CEP**	-	-	-	-	98.9	1.1	0.0	0.0	-	-	-	-	-	2	2
**XNL**	-	-	75	23.9	1.1	0.0	-	-	-	-	-	-	-	0.5	1
**SDM**	-	-	-	-	-	-	-	-	2.2	0.0	1.1	9.8	87	>256	>256
**P/N**	-	-	-	100	0.0	0.0	0.0	-	-	-	-	-	-	1	1

**Legend:** PEN-penicillin, AMP-ampicillin, ERY-erythromycin, Oxa+-oxacillin + 2%NaCl, RIRL–**pirlimycin**, TET-tetracycline, CEP–cephalothin, XNL-ceftiofur, SDM–sulphadimethoxine, P/N–penicillin/novobiocin. Susceptibility breakpoints are highlighted in light blue whereas resistance breakpoints: in dark blue (CLSI, 2018). Intermediate isolates were interpreted as resistant. NI = the growth of the isolate is not inhibited by the highest concentration of the antimicrobial agent in the plate. MIC_50_ (μg/mL) = concentration inhibiting 50% of isolates. MIC_90_ (μg/mL) = concentration inhibiting 90% of isolates.

**Table 2 vetsci-09-00401-t002:** Comparison between disk diffusion and minimum inhibitory concentration methods for testing resistance of bovine mastitis *S. aureus* isolates to 9 chemotherapeutic drugs (penicillin, ampicillin, cefoxitin/oxacillin, cephalotin, ceftiofur, erythromycin, tetracycline, penicillin/novobiocin, pirlimycin).

	Minimum Inhibitory Concentration		Total
	N Sensitive Isolates	N Resistant Isolates	
**Disk diffusion test** **N sensitive isolates**	55	5	60
**Disk diffusion test** **N resistant isolates**	2	30	32
**Total**	57	35	92

## Data Availability

The data presented in this study can be found in the manuscript.

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
