# Peer review of "Comparison between Some Phenotypic and Genotypic Methods for Assessment of Antimicrobial Resistance Trend of Bovine Mastitis Staphylococcus aureus Isolates from Bulgaria"

_vetsci, 2022, doi:10.3390/vetsci9080401_

Round 1

Reviewer 1 Report

I express my deep appreciation and respect for the work done by the authors of this article. This paper presents interesting and original results of experimental studies. I read the manuscript quickly and easily, all the results are presented clearly. The results in the article are presented in such a way that the article can be understood by a wide range of people, not only specialists in the field of surgical dentistry. 

I recommend to publish this article in present form. 

Reviewer 2 Report

Comments:

·      In Material Methods section 2.2 include the primer sequence and the agarose gel percentage used for PCR.

·      In result section 3.1 include the identification results in either main figure or supplementary figure.

·      In result section 3.2 include positive control in gel run, this will further validate the product size.

·      In result section 3.3 provide supplementary figures of the disk diffusion experiment

·      In section 3.4 in figure legend write Ðœ – 100 bp write the company name of the ladder as it is mentioned in figure 1, also in figure 1 the negative control is written as NTC.So, please maintain the same labelling pattern.If possible include positive control

·      In section 3.4 line 294-295 first write 310 bp and then 517 bp means write the way it was loaded on gel also include the percentage of gel

Reviewer 3 Report

About this study:

Authors evaluated the resistance of bovine mastitis Staphylococcus aureus isolated from 14 Bulgarian farms, by means of disk diffusion / minimum inhibitory concentrations tests for seven most used antibiotics, to investigate the prevalence of main genes encoding antimicrobial resistance and to compare phenotypic and genotypic tests using statistic means.

Hardpoints:

Introduction: is well written introducing the manuscript reader to the topic

M & M: is clearly described and easy replicable

 Discussion:  

What co improve:

The title as it is does not cover entirely the Phenotypic and Genotypic comparative aspects.

Figure 2 is not statistically interpreted by comparing the results from all farms the values of p not being provided. A representation with T-bars from ANOVA would be recommendable. Also, Figure 2 and Table 3 are not necessarily linked with the current research. In Table are not indicate the authors who determined MIC concentrations.

S. aureus mechanisms of resistance to antibiotics are well acknowledged in general. The blaZ gene, for β-lactamases,  mecA gene, encoding, ermC genes, and tetK genes are presented also here, but for a complete view/comparison, the studied structures have to be seven. So, if you can provide more information about other involved genes in resistance to the cephalosporins group, for example, would be excellent. Stress more this part.

Decision: major revision

Reviewer 4 Report

First: The manuscript needs a significant linguistic improvement. It is badly written and the bad language makes reading and understanding difficult.

Introduction. The introduction is longer than necessary. Some passages with well-known facts are redundant and can be deleted.

The last paragraph, with the objectives of the study, must be written more clearly.

Materials and methods

2.1 Please explain the selection procedures for the animals and the samples. Did you have some inclusion / exclusion criteria? If yes, please list, if no, please justify.

If this was convenience sampling, please include a passage in Discussion to debate the limitations.

The identification of staphylococci was erratic, this is a significant error, and must be corrected.

2.2. Please present all the details of the PCRs in supplementary material. The primers must be presented in minute detail, the conditions of the PCR (temperature, product size etc.) must be described.

2.3. Which criteria were employed for phenotypic assessment of resistance? Please described.

Results

Table 3 is too detailed and should be transferred to supplementary material.

Same for tables 4-8. Only a summary of these results must be presented in the main text.

Discussion

This is very long and verbose for some clear-cut findings. Please reduce in length and please avoid repetitive text.

Overall. The points mentioned above must be corrected. The identification of staphylococci has not been described convincingly. The manuscript is very long for a generally brief story.

Major revision with significant shortening and resubmission as communication.

Round 2

Reviewer 3 Report

The authors tried to do their best to improve the manuscript.

At this stage, the manuscript is acceptable in my eyes.

Author Response

Dear reviewer,

Thank you very much for accepting our paper. We did our best to perform the research, to write and revise it. 

Reviewer 4 Report

Section 1. The introduction is still very long. Please delete some passages, especially those with well-known facts.

Sub-section 2.2. Still, I am not happy with the details of the PCR. Please include all these in a table and please move to supplementary material.

Table 1 should be used to supplementary material.

Sub-section 2.3. Why using the CLSI criteria, when there are the EUCAST standards? This is a serious mistake for a European work. Please go back to the raw data and recalculate everything by using the EUCAST norms. Then, re-assess the work in view of the revised findings.

Table 2 misses the annealing temperatures. Also, the table must be transferred to supplementary material.

Significant changes must be implemented and further details assessment is necessary. Lack of application of European-based criteria is a serious flaw, but can be corrected.
